# The role of the mitochondrial outer membrane protein SLC25A46 in mitochondrial fission and fusion

Jana Schuettpelz , Alexandre Janer , Hana Antonicka , Eric A Shoubridge 

**Mutations in *SLC25A46* underlie a wide spectrum of neurodegenerative diseases associated with alterations in mitochondrial morphology. We established an SLC25A46 knock-out cell line in human fibroblasts and studied the pathogenicity of three variants (p.T142I, p.R257Q, and p.E335D). Mitochondria were fragmented in the knock-out cell line and hyperfused in all pathogenic variants. The loss of SLC25A46 led to abnormalities in the mitochondrial cristae ultrastructure that were not rescued by the expression of the variants. SLC25A46 was present in discrete puncta at mitochondrial branch points and tips of mitochondrial tubules, co-localizing with DRP1 and OPA1. Virtually, all fission/ fusion events were demarcated by a SLC25A46 focus. SLC25A46 co-immunoprecipitated with the fusion machinery, and loss of function altered the oligomerization state of OPA1 and MFN2. Proximity interaction mapping identified components of the ER membrane, lipid transfer proteins, and mitochondrial outer membrane proteins, indicating that it is present at interorganellar contact sites. SLC25A46 loss of function led to altered mitochondrial lipid composition, suggesting that it may facilitate interorganellar lipid flux or play a role in membrane remodeling associated with mitochondrial fusion and fission.**

## Introduction

Mitochondria are dynamic organelles whose morphology responds to physiological signals and metabolic stresses by altering the balance between organelle fission and fusion. ATP production by oxidative phosphorylation (OXPHOS) is a core mitochondrial function, but mitochondria also play crucial roles in calcium signaling, the biosynthesis of phospholipids, iron–sulfur clusters, and heme, and they serve as platforms for the innate immune response. Mitochondrial fission is necessary for the cellular distribution of mitochondria and their segregation into daughter cells during mitosis, and fission is also thought to be important for the initiation of apoptosis, for the turnover of damaged organelles by mitophagy, and for the innate immune response (Kraus et al, 2021; Tabara et al, 2021). Mitochondrial fusion is thought to be a response to physiological stress and to provide a mechanism to rescue depolarized organelles and prevent their turnover, and potentially to mix organelle contents (Tilokani et al, 2018; Mattie et al, 2019).

Mitochondrial fission requires the recruitment of the cytosolic large molecular weight GTPase DRP1 to adapter proteins (MFF, MID49, MID51, and FIS1) at the outer membrane (Loson et al, 2013), whereas fusion is orchestrated by the conserved large molecular weight GTPases MFN1, MFN2, and OPA1 (Song et al, 2009). Mutations in *MFN2* cause Charcot–Marie–Tooth type 2, an axonal peripheral neuropathy (Zuchner, 1993). Mutations in *OPA1*, depending on the allelic variant, are associated with a variety of diseases. Although haploinsufficient-related mutations in *OPA1* lead to optic atrophy, dominant gain-of-function mutations in *OPA1* cause more severe phenotypes of optic atrophy, often concomitant with deafness, axonal neuropathy, ataxia, and/or myopathy. Compound-heterozygous mutations lead to the Behr syndrome, illustrating the genotypic complexity (Chao de la Barca et al, 2016).

In yeast, Ugo1p coordinates Fzo1p (homolog of MFN1/2) dimerization, protecting it from degradation (Anton et al, 2011), and it links Fzo1p at the outer membrane with Mgm1p (homolog of OPA1) at the inner membrane (Hoppins et al, 2009). How this complex sequence of events occurs in mammals remains largely unknown.

SLC25A46 is the mammalian homolog of Ugo1p (Abrams et al, 2015; Janer et al, 2016), suggesting that it could play a similar role in coordinating mitochondrial fusion. SLC25A46 is a member of the SLC25 family of mitochondrial metabolite carriers that comprises 53 proteins in humans, about two thirds of which have been functionally characterized (Palmieri et al, 2020). Most of these carry small metabolites across the inner mitochondrial membrane. All SLC25 family members are integral membrane proteins with six transmembrane helices, and the majority have a highly conserved consensus sequence—PX(D,E)XX(K,R)—at the C-terminal end of the odd-numbered helices in all three repeats (Kunji et al, 2020). The charged residues in this sequence form salt bridges that close the pore, through which metabolites are translocated, on the matrix side of the protein (Monne et al, 2013). This consensus sequence is not conserved in SLC25A46, and as SLC25A46 localizes to the outer mitochondrial membrane (OMM) (Abrams et al, 2015; Janer et al,

Department of Human Genetics, Montreal Neurological Institute, McGill University, Montreal, Canada

Correspondence: eric.shoubridge@mcgill.ca

2016; Wan et al, 2016), which is permeable to small molecules up to 5 kD through VDAC (porin) channels (Camara et al, 2017), it seems unlikely that SLC25A46 functions as a conventional metabolite transporter.

The first mutations in *SLC25A46* were reported in four patients with biallelic loss-of-function mutations who presented with cerebellar atrophy, optic atrophy, and Charcot–Marie–Tooth type 2 (Abrams et al, 2015). Using whole-exome sequencing, we uncovered a homozygous missense mutation in *SLC25A46*, in a patient with the Leigh syndrome, an early-onset, fatal neurodegenerative disorder (Janer et al, 2016). Subsequently, 29 patients from 22 families were reported with biallelic missense mutations in *SLC25A46* (Fig 1A) in an increasingly broad spectrum of neurological disorders that includes progressive myoclonic ataxia, autosomal recessive cerebellar ataxias, pontocerebellar hypoplasia with spinal muscular atrophy (PCH1), and Parkinson's disease and optic atrophy (Charlesworth et al, 2016; Wan et al, 2016; Hammer et al, 2017; Nguyen et al, 2017; Sulaiman et al, 2017; van Dijk et al, 2017; Abrams et al, 2018; Braunisch et al, 2018; Bitetto et al, 2020; Ababneh et al, 2021; Kodal et al, 2022). The onset and course of the disease are highly variable, ranging from fetal death to survival to 50 yr of age. Almost all pathogenic mutations predict amino acid substitutions within the IMS-facing loops of the protein (Fig 1A), suggesting a role of these sites either in protein–protein interactions or more likely in targeting and insertion of the protein to the OMM (the mechanism of which remains unknown). Disease severity is inversely correlated with the steady-state levels of SLC25A46, suggesting that all pathogenic variants are hypomorphs (Abrams et al, 2018).

In this study, we investigated the role of SLC25A46 in the control of mitochondrial morphology and lipid homeostasis by first knocking out SLC25A46 in fibroblasts and subsequently expressing the WT

cDNA or the cDNA of three different pathogenic variants spanning the range of protein stability and disease severity (Fig 1B and C).

# Results

## Mitochondrial fragmentation in SLC25A46 knock-out cells and hyperfusion in pathogenic variants

To investigate the molecular basis for SLC25A46 pathogenicity, we first established a knock-out in a control immortalized human fibroblast cell line using CRISPR/Cas9 editing (Fig 1B). We then created stable cell lines expressing either WT SLC25A46 or one of the three different pathogenic variants (p.T142I, p.R257Q, and p.E335D) that span the range of phenotypic severity (Fig 1B). The expression of the individual SLC25A46 pathogenic variants did not reach WT levels, confirming a previous study showing a negative correlation between residual steady-state protein levels and the age of onset and severity of disease (Abrams et al, 2018) (Figs 1C and S1). The knock-out cell line had a proliferation defect that was rescued by the expression of the WT protein, but not by the expression of the pathogenic variants (Fig 1D).

We next examined mitochondrial morphology in these cell lines as previous studies reported that the loss of the SLC25A46 function led to mitochondrial hyperfusion (Abrams et al, 2015; Janer et al, 2016; Steffen et al, 2017). The complete loss of SLC25A46 in the knock-out resulted in a marked fragmentation of the mitochondrial network (Fig 2A and B). These findings were confirmed by creating a CRISPR/Cas9-induced SLC25A46 knock-out cell line in HeLa cells (Fig S2A), which contrasts with previous results of acute siRNA–mediated knock-down (Janer et al, 2016). Extensive mitochondrial

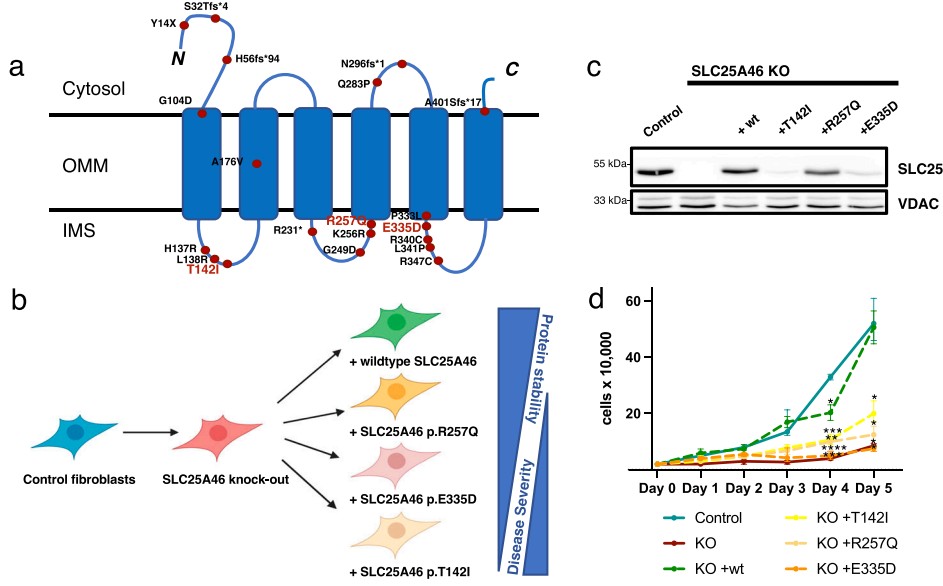

**Figure 1. Pathogenic mutations in SLC25A46 and the cell lines created for this study.**
**(A)** Schematic representation of the outer mitochondrial membrane protein SLC25A46 showing the known pathogenic variants. The predicted six transmembrane domains (UniProt) are indicated. Most pathogenic variants occur in the loops between the transmembrane domains facing the intermembrane space. The pathogenic variants that we have chosen to study are indicated in red letters. **(B)** Knock-out human fibroblast line was created using CRISPR/Cas9 editing and rescued either with the WT protein or with a pathogenic variant reintroduced using a retroviral expression vector: p.T142I (Leigh syndrome) (Janer et al, 2016), p.R257Q (optic atrophy, axonal neuropathy, and ataxia) (Abrams et al, 2018), and p.E335D (sensory and motor neuropathy) (Abrams et al, 2015). **(C)** SDS–PAGE analysis of control human fibroblasts, SLC25A46 knock-out fibroblasts, and knock-out fibroblasts overexpressing the WT SLC25A46 or a pathogenic variant (T142I, R257Q, or E335D). VDAC serves as a loading control. **(D)** Proliferation assay of the indicated fibroblast cell lines. Statistical analysis was done by two-way ANOVA compared with the control cell line; data are shown as the mean + SEM (n = 3).
Source data are available for this figure.

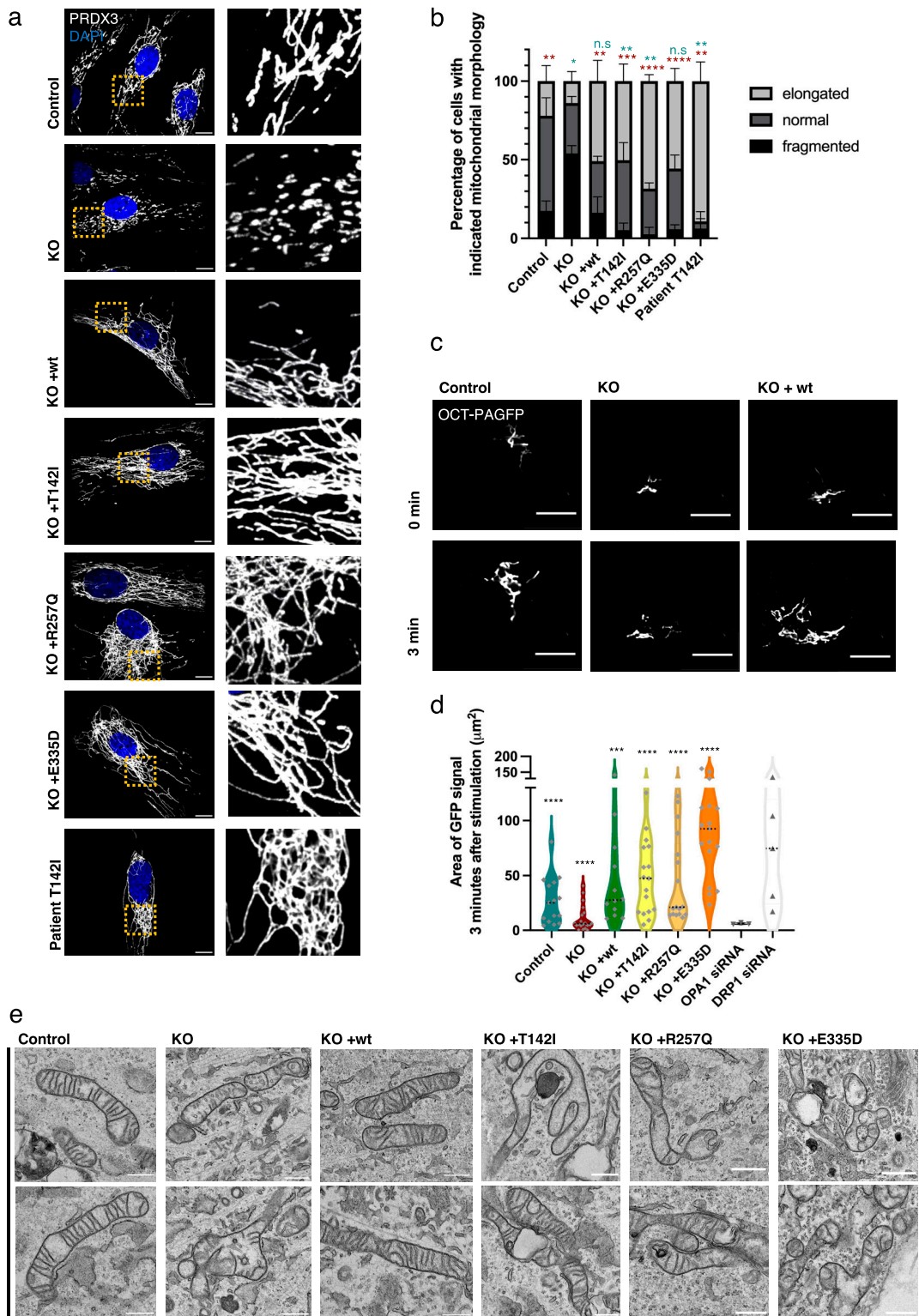

**Figure 2.   Mitochondrial fragmentation in SLC25A46 knock-out cells and hyperfusion in pathogenic variants.**
**(A)** Mitochondrial morphology of control fibroblasts, the knock-out cell line, cells with the reintroduced SLC25A46 protein (WT, p.T142I, p.R257Q, and p.E335D), and the Leigh syndrome patient cell line (p.T142I) was analyzed by immunofluorescence. Representative images of fibroblasts decorated with anti-PRDX3 (mitochondria) in white and DAPI (nuclei) in blue. A zoomed image of the indicated area is shown on the right. Scale bars: 10 $\mu$m. **(B)** Quantification of cells according to their mitochondrial network morphology (randomized, n > 60 cells per condition, N = 3; data are shown as the mean + SEM). Statistical analysis was done using an unpaired $t$ test. Blue stars indicate the $t$ test analysis fraction of elongated cells from the total amount of cells compared with the control cell line. Red stars indicate $t$ test analysis fraction of fragmented

hyperfusion was observed in cells expressing the pathogenic variants and in the patient fibroblast cell line (Fig 2A and B), as previously described (Janer et al, 2016).

To confirm and extend these findings, we quantified mitochondrial interconnectivity by expressing a photoactivatable GFP probe targeted to the mitochondrial matrix (OCT-PAGFP) (Patterson & Lippincott-Schwartz, 2002). As a validation for the method, we treated control fibroblasts with OPA1 siRNA or DRP1 siRNA, which, as expected, led to decreased or increased interconnectivity, respectively. Consistent with the morphological findings, the knock-out of SLC25A46 resulted in reduced mitochondrial interconnectivity and the expression of the pathogenic variants led to increased mitochondrial interconnectivity (Figs 2C and D and S2B). Transmission electron microscopy (TEM) demonstrated severely impaired mitochondrial cristae morphology in SLC25A46 knock-out cells and in cells expressing the pathogenic variants. Mitochondria were swollen, and cristae were either not present or not aligned (Fig 2E). The ultrastructural defects in the knock-out cells were rescued by re-expressing WT SLC25A46. Although the cristae architecture was highly compromised in the knock-out or pathogenic variant–expressing cells, surprisingly assembly of OXPHOS complexes as assessed by BN-PAGE was not compromised (Fig S2C).

### SLC25A46 localizes to foci at mitochondrial tips and branches and is present at fusion and fission sites

The clear morphological changes in the mitochondrial network in the SLC25A46 knock-out cell and in cells with pathogenic variants suggest that SLC25A46 may play a role in both mitochondrial fusion and fission. To determine whether SLC25A46 is an active player in these processes, we examined the sub-organellar localization of SLC25A46 by expressing SLC25A46-GFP in fibroblasts. Our previous study showed that the C-terminally GFP-tagged protein is functional, as it was able to rescue the cellular phenotypes in SLC25A46 patient fibroblasts (Janer et al, 2016). Immunofluorescence experiments showed that SLC25A46-GFP is focally expressed at most mitochondrial tips and branch points (80%) (Fig 3A and B). To ensure that the localization of SLC25A46-positive foci was not an effect of the overexpression of the protein with a GFP tag, we analyzed endogenous levels of SLC25A46 in fibroblasts by immunofluorescence with an anti-SLC25A46 antibody (Fig S3A and B). Endogenous SLC25A46 was also present in foci at nearly all mitochondrial tips (97%) and branches (94%). SLC25A46 was also focally expressed, often at the tips of mitochondria, in cerebral neurons differentiated from human induced pluripotent stem cells (Fig S3C). We were able to analyze the localization of the highest expressing pathogenic variant R257Q and confirmed that it is also focally present at mitochondrial tips and branches (Fig S3D and E).

We next used live-cell imaging to examine whether SLC25A46 was involved in mitochondrial fusion/fission events. This analysis demonstrated that SLC25A46 is present at virtually all

mitochondrial fusion and fission sites (Fig 3C and D and Video 1, Video 2, and Video 3), suggesting an active role in these processes.

### Association of SLC25A46 with the proteins of the fusion and fission machinery

Analysis of the protein proximity interaction network of SLC25A46 (Antonicka et al, 2020) indicated that SLC25A46 interacts with both the mitochondrial fusion and fission machinery (Fig 4A and Table S1) and with proteins involved in intracellular vesicular transport. Among the most specific proximity interactors of SLC25A46 compared with all other mitochondrial proteins investigated in the study (Antonicka et al, 2020) were proteins involved in vesicle budding from the membrane and proteins involved in organelle organization. We focused on the investigation of mitochondrial proximity interactors of SLC25A46. We performed immunoprecipitation of endogenous SLC25A46, followed by immunoblotting (Fig 4B). SLC25A46 co-immunoprecipitated with MFN1, MFN2, and OPA1, proteins of the mitochondrial fusion machinery, and one of the proteins of the MICOS complex (MIC25), in agreement with previously reported data (Janer et al, 2016). Reciprocal immunoprecipitation of the OPA1 protein was able to co-immunoprecipitate other proteins of the fusion machinery and SLC25A46, indicating the proximity of these two proteins in mitochondria. Immunofluorescence analysis showed that SLC25A46-positive punctae are in proximity to foci of not only the fusion protein OPA1, but also the fission protein DRP1 (DNM1L) (Fig 4C), a proximity interactor of SLC25A46 (Fig 4A). Stimulated emission depletion (STED) microscopy analysis further demonstrated that SLC25A46 surrounds the OPA1 punctae as expected (Fig 4D), as the proteins do not share the same membrane. These data indicate a close relationship between SLC25A46 and mitochondrial fusion and fission machinery.

### MFN2 high molecular oligomeric complexes are decreased in SLC25A46 loss-of-function cells

As mitochondrial morphology is altered in SLC25A46 knock-out cells and in cells with pathogenic variants, we next investigated the fusion and fission machinery in these cells (Fig 5A and B). No substantive differences in the steady-state protein levels of MFN1/2, OPA1, or DRP1 were detected on conventional immunoblots, which contrasts with the findings by Steffen et al (2017) who showed an alteration in MFN1 and MFN2 protein expression in HEK293T SLC25A46 knock-out cells (Figs 5A and S4A). Steffen et al reported a reduced half-life of the pathogenic variant L341P, and we confirmed a reduction in the half-life of the three pathogenic variants we studied (Fig S4B and C). To test whether the pathogenic variants are inserted into the OMM, we performed a PK assay on the most abundant variant R257Q and an alkaline carbonate extraction assay on all pathogenic variants (T142I, R257Q, and E335D) (Fig S5A and B).

cells from total cells compared with knock-out. **(C)** Live-cell imaging analysis of fibroblasts expressing a mitochondrial-targeted photoactivatable GFP (OCT-PAGFP) probe. Representative images of the OCT-PAGFP probe diffusion over a 3-min period after activation with the 405-nm laser. Scale bars: 10 $\mu$m. **(D)** Quantification of the GFP signal area 3 min after stimulation for each cell type. Statistical analysis was done by a Wilcoxon signed-rank test. Data represent n > 10. **(E)** Representative TEM images of the indicated fibroblast cells showing impaired mitochondrial cristae morphology in KO cells and in cells with pathogenic variants. Scale bars: 800 nm.

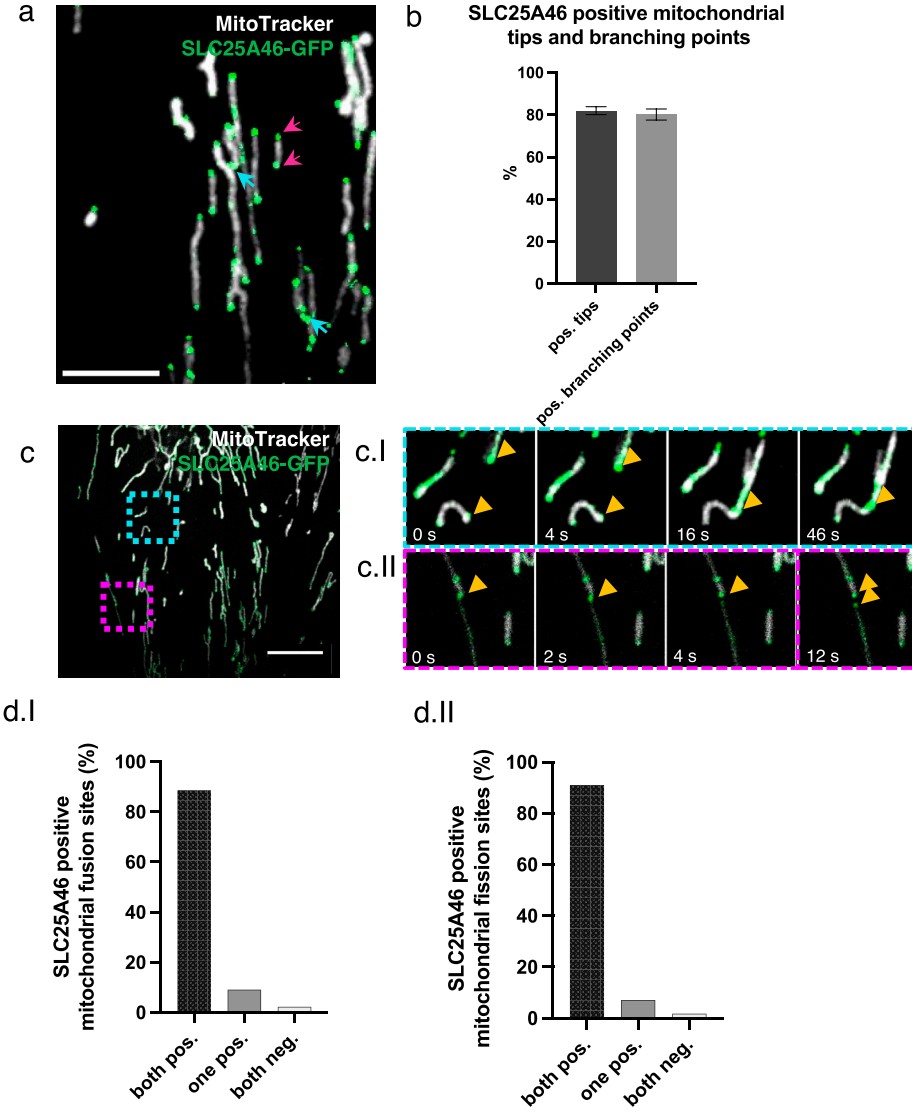

**Figure 3. SLC25A46 localizes to foci at mitochondrial tips and branches and is present at fusion and fission sites.**
**(A)** Fibroblasts stably overexpressing SLC25A46-GFP (green) were analyzed by immunofluorescence. Mitochondria were stained with MitoTracker Deep Red (white). Pink arrows indicate mitochondrial tips. Blue arrows indicate mitochondrial branch points. Scale bar: 5 µm. **(B)** Mitochondrial tips (n > 70 per condition) and branching points (n > 20 per condition) were analyzed for focal SLC25A46-GFP signal. Data are shown as the mean + SEM, n = 3. **(C)** Fibroblasts stably overexpressing SLC25A46-GFP (green) were analyzed in live-cell imaging. Mitochondria were stained with MitoTracker Deep Red (white). Images were captured every 0.5 s for a period of 1 min. Scale bar: 10 µm. (ci) Time-lapse imaging of a fusion event happening in the blue box. (cii) Time-lapse imaging of a fission event happening in the pink box. **(D)** Quantification of the SLC25A46-GFP–positive fusion (di) and fission (dii) events. Events were analyzed for the presence of focal SLC25A46-GFP–positive signal at the tip of the two parental mitochondria before fusion (di) and at the tip of the two daughter mitochondria after fission (dii). Events were classified as follows: both positive, when a signal was present at both mitochondrial tips; one positive, when a signal was present at only one mitochondrial tip; or both negative. 10 cells were analyzed with a total of 44 fusion and 56 fusion events, respectively.

The PK assay showed that the R257Q variant behaves as the WT protein and as the outer membrane protein MFN2 (Fig S5A). The alkaline carbonate assay showed that all pathogenic variants (T142I, R257Q, and E335D) are integral membrane proteins (Fig S5B).

The SLC25A46 homolog Ugo1p in yeast has been reported to be required for the complex assembly of the MFN1/2 homolog Fzo1p and the OPA1 homolog Mgm1p (Hoppins et al, 2009). As steady-state levels of these proteins were not affected in our model, we investigated whether the assembly of MFN1/2 and OPA1 complexes was changed in SLC25A46 knock-out cells and cells expressing the pathogenic variants. Isolated mitochondria were solubilized with 1% digitonin and separated on Blue-Native PAGE (BN-PAGE), and individual complexes were visualized by specific antibodies (Fig 5B). In control cells, MFN2 and OPA1 migrated predominantly as either oligomers or monomers, whereas MFN1 ran as a monomer (Fig 5B). In SLC25A46 knock-out cells and in the pathogenic variants, the complex formation of either MFN2 or OPA1, but not MFN1, was affected (Fig 5B). The oligomerization of MFN2 was dependent on

the steady-state level of SLC25A46 expression (Fig 5C). The knock-out of SLC25A46 led to reduced MFN2 complex formation, which was rescued by the overexpression of either WT or variants. MFN2 oligomerization correlated with the expression level of SLC25A46 (Fig 5C), suggesting that the steady-state level of the SLC25A46 protein is an important determinant of the underlying defect.

### OPA1 high molecular weight oligomeric complexes are increased, and their forms are altered in SLC25A46 loss-of-function cells

OPA1 has multiple isoforms, which can be further processed by the two proteases OMA1 and YME1L1. The long, membrane-bound forms are required for the mitochondrial inner membrane fusion, whereas the short, soluble forms are associated with inhibiting fusion and promoting fission (MacVicar & Langer, 2016). BN-PAGE analysis showed a slight increase in OPA1 high molecular weight complex formation in the SLC25A46 knock-out cell line compared

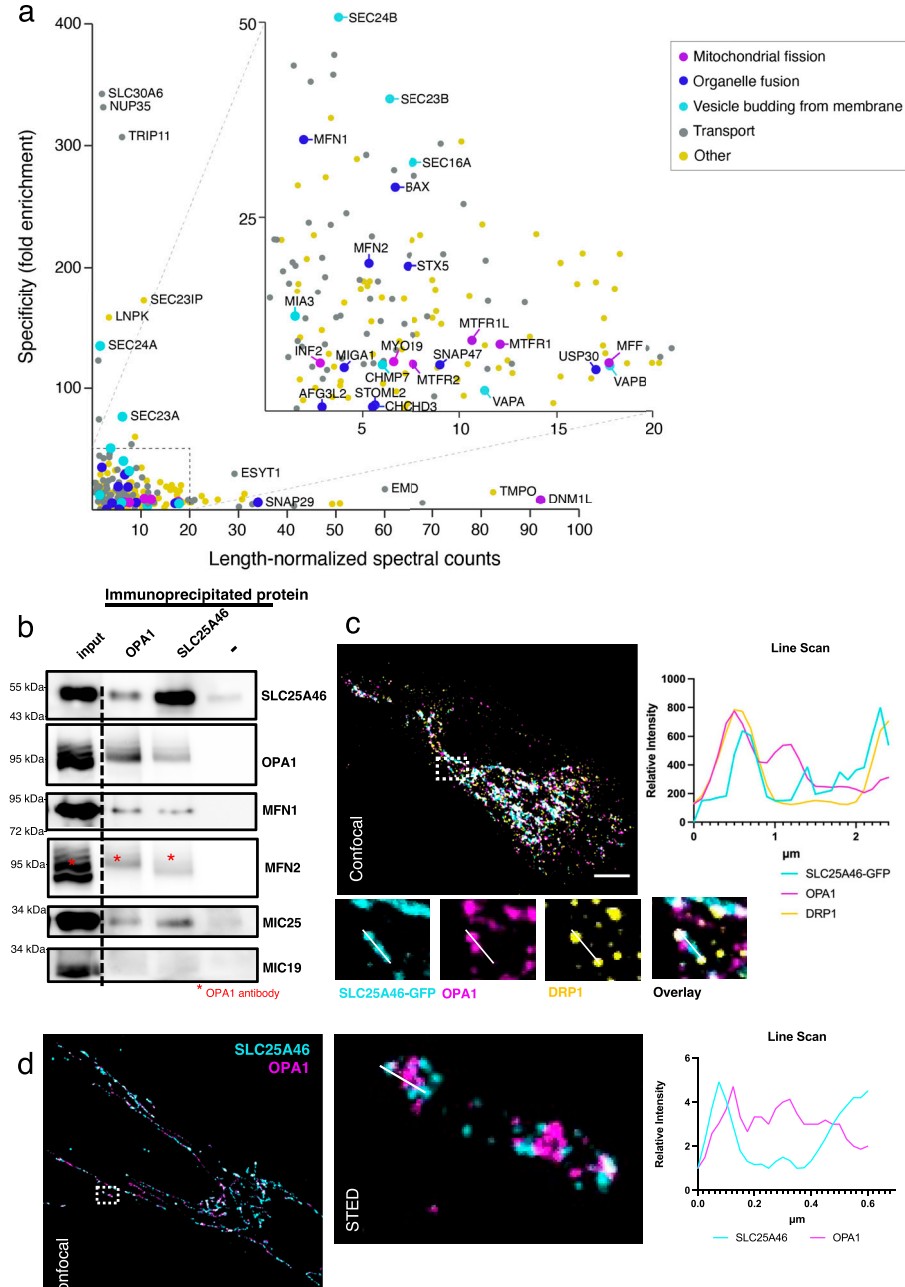

**Figure 4. Association of SLC25A46 with the proteins of the fusion and fission machinery.**
**(A)** Specificity plot of SLC25A46 (N-terminally BirA*-tagged) indicating the specificity (defined as fold enrichment) of interaction between preys identified as proximity interactors of SLC25A46 compared with their interaction with all the other baits in the dataset (Antonicka et al, 2020). GO term analysis was performed using g:Profiler, and proteins belonging to the indicated GO:BP terms are highlighted. **(B)** Endogenous SLC25A46 and OPA1 were immunoprecipitated from human fibroblast crude mitochondria, and co-immunoprecipitated proteins were separated on SDS–PAGE. Western blot analysis with indicated antibodies was performed. **(C)** Confocal microscopy analysis of fibroblasts expressing GFP-tagged SLC25A46. SLC25A46 is shown in cyan, OPA1 in magenta, and DRP1 in yellow. A line scan of the specified line in the image indicates the intensity of the signal of SLC25A46 in cyan, of OPA1 in magenta, and of DRP1 in yellow. Scale bar: 10 μm. **(D)** Stimulated emission depletion microscopy analysis of human fibroblasts. Endogenous SLC25A46 (in cyan) and OPA1 (in magenta) were detected by confocal (left) microscopy and stimulated emission depletion (right, a zoom of the boxed region in the confocal image). Scale bar: 10 μm. A line scan of the specified line in the image indicates the intensity of the signal of SLC25A46 in cyan and of OPA1 in magenta.

with control fibroblasts (Fig 5B, green and blue box). Because BN-PAGE is unable to distinguish between the individual isoforms, we performed a two-dimensional PAGE (BN-PAGE/SDS–PAGE) analysis to separate the long and short forms (Fig 6A). In control cells, both long and short forms of OPA1 were present as monomers (Fig 6A, red box), whereas long forms were predominantly found in the higher molecular weight complex (~200–400 kD, blue box). In the knock-out of SLC25A46, the long and short forms were also present as monomers and in a 200- to 400-kD complex (Fig 6A, blue box), but to a lesser extent. Interestingly, OPA1 was also present in even larger complexes (>600 kD), where the short forms were more

prominent (Fig 6A [green box], Fig 6B–D). Cells re-expressing the WT form of SLC25A46 present a similar OPA1 distribution pattern to the control cells, demonstrating the specificity of this phenotype. A similar pattern for OPA1 complex formation as in the KO cell line (the presence of the >600-kD complex) was detected in the cell line expressing the pathogenic variant p.T142I; however, the long OPA1 forms were predominantly present in the high molecular weight complexes (Fig 6A, green box and blue box, Fig 6B–D). Thus, the presence of short forms in the knock-out and long forms in the pathogenic variant p.T142I in >600-kD complexes (green box) correlates with the morphology observed in these cells, fragmentation

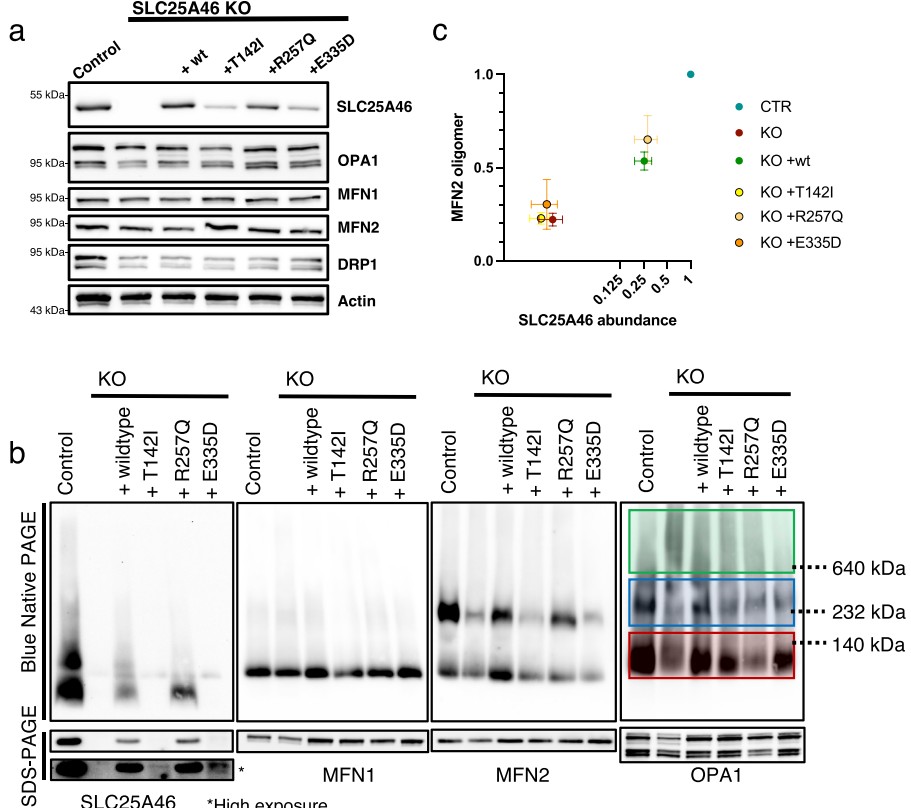

**Figure 5. MFN2 high molecular weight oligomeric complexes are decreased in SLC25A46 loss-of-function cells.**
**(A)** SDS–PAGE analysis of steady-state levels of fusion and fission machinery proteins in control fibroblasts, SLC25A46 knock-out fibroblasts, and knock-out fibroblasts overexpressing the WT SLC25A46 or pathogenic variants (T142I, R257Q, and E335D). Actin serves as a loading control. **(B)** BN-PAGE and SDS–PAGE analysis of 1% digitonin extracts from crude mitochondria prepared from indicated cell lines. Membranes were immunoblotted for the indicated proteins. OPA1 complexes are highlighted: green boxes indicate the high molecular weight complexes (>600 kD), blue boxes indicate the molecular weight complexes of 200–400 kD, and red boxes indicate the monomeric OPA1 forms. **(C)** Quantification of the MFN2 higher molecular weight complex in BN-PAGE (normalized to control) presented as a ratio to SLC25A46 (normalized to control). SLC25A46 levels are presented in $\log_2$ to visualize the lower values. n = 3 independent experiments; data are shown as the mean + SEM.
Source data are available for this figure.

versus hyperfusion (Fig 2A). 2D gels also confirmed our observation from BN-PAGE that oligomers of MFN2 are absent in KO and p.T142I cells.

## MFN2 oligomerization is influenced by SLC25A46 and OPA1

As SLC25A46 plays a role in the formation of MFN2 and OPA1 oligomers, we investigated the consequence of MFN2 or OPA1 knock-down in the cells lacking SLC25A46 (Fig 7A). Knock-down of OPA1 resulted in an increase of MFN2 steady-state levels in both the control and SLC25A46 knock-out cells (Fig 7A, SDS–PAGE), and this increase was sufficient to rescue the level of MFN2 complex formation in the SLC25A46 knock-out cell line, suggesting that the oligomeric complex of MFN2 is a compensatory response to the loss of OPA1 function. This was not, however, sufficient to alter mitochondrial morphology, as OPA1 knock-down resulted in mitochondrial fragmentation in both control and KO cells (Fig 7B). As expected, we detected a disassembly of OXPHOS complex V when OPA1 was silenced, because complex V requires adequate cristae morphology and therefore is dependent on OPA1 (Cogliati et al, 2013). MFN2 knock-down induced the disassembly of complex V in the absence of SLC25A46 (Fig 7A) and a concomitant fragmentation of the mitochondrial network (Fig 7B). The effect of the MFN2 knock-down in the SLC25A46 knock-out cells compared with control fibroblasts (Fig 7B) indicates a clear dependence of mitochondrial morphology on the presence of SLC25A46.

## SLC25A46-dependent changes in the mitochondrial lipid profile

The oligomerization of GTPases and the curvature of the membranes during fusion and fission events are highly dependent on the lipid content of the mitochondrial membrane (Low et al, 2009; Frohman, 2015; Brandt et al, 2016; Brandner et al, 2019). Our previous results showed an alteration in mitochondrial lipid content in a Leigh syndrome patient with the p.T142I mutation (Janer et al, 2016). To test whether the alteration in mitochondrial morphology, cristae structure, and oligomerization of MFN2 and OPA1 are due to an altered lipid content, we performed shotgun mass spectrometry lipidomics on sucrose bilayer purified mitochondria from the control, knock-out, WT-rescue, and the pathogenic variant p.T142I, allowing broad coverage of lipids and absolute quantification (Lipotype GmbH).

From 1,798 identified lipid entities, we found 203 lipids that were significantly different (0.05%) with a fold change of >2 Table S2). Using multi-variant data analysis by principal component analysis, we showed the clustering of the samples of the same cohorts, demonstrating the quality of the data, and the separation of the different cohorts, suggesting a different mitochondrial lipid content (Fig 8A). Hierarchical clustering with heatmap analysis of lipid classes showed a clear difference in quantities of most classes of lipids between control and knock-out mitochondria, which was rescued by the expression of the WT variant but not by the expression of the pathogenic variant T142 (Figs 8B and C and S6 and

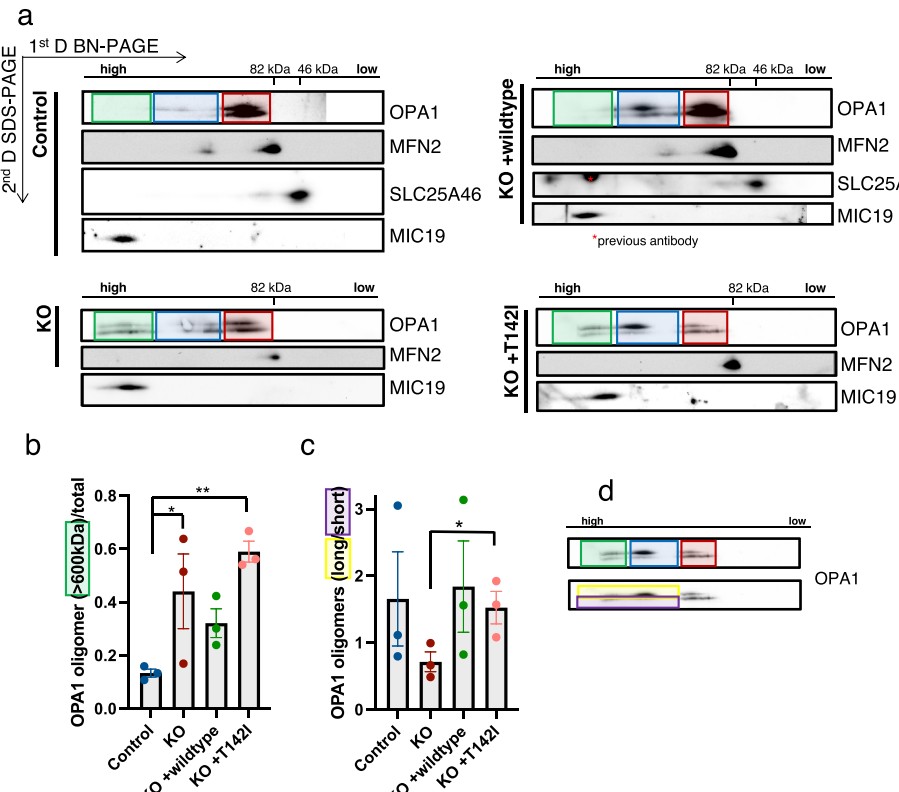

**Figure 6. OPA1 high molecular oligomeric complexes are increased, and the short versus long forms are altered in SLC25A46 loss-of-function cells.**
**(A)** Two-dimensional electrophoresis (BN-PAGE/SDS–PAGE) analysis of mitochondria from the control, knock-out, and re-expression of WT-SLC25A46 and the pathogenic variant p.T142I. Membranes were immunoblotted for the indicated proteins. OPA1 complexes are highlighted: green boxes indicate the high molecular weight complexes (>600 kD), blue boxes indicate the molecular weight complexes of 200–400 kD, and red boxes indicate the monomeric OPA1 forms. **(A, B)** Quantification of the high molecular weight complexes (>600 kD) of OPA1 (indicated in the green boxes in (A)) relative to the total signal (n = 3). **(C, D)** Quantification of the relative proportions of the long versus short forms of OPA1 forms in the high molecular weight complexes (>200 kD) as indicated in the example (D) showing the longer forms of the higher complexes in the yellow box and the shorter forms in the purple box (n = 3).

Table S2). The knock-out of SLC25A46 resulted in a decrease in the three main mitochondrial lipid categories, cardiolipin (CL), phosphatidylethanolamine (PE), and phosphatidylinositol (PI), and an increase in phosphatidylcholine (PC) (Figs 8B and C and S6). Although PC increased in the SLC25A46 knock-out cell line, the ether-linked PC (PC O-) was decreased. In contrast, the decrease in PE was accompanied by an increase in ether-linked PE (PE O-) (Figs 8B and C and S6 and Table S2). CL was overall decreased in the knock-out cell line compared with control fibroblasts, and more specifically, we observed individual changes, some subclasses were decreased (CL 68:4; 0, CL 70:4;0, and CL 72:5;0), whereas others were increased in the knock-out (CL 72:6;0 and CL 72:7;0) (Fig 8D and Table S2). All alterations in the lipid profiles of the knock-out cells were rescued by the expression of the WT protein but not the pathogenic variant (Figs 8 and S6 and Table S2). Interestingly, the levels of 17 different lipid subspecies were significantly different between the knock-out cells and cells expressing the p.T142I pathogenic variant (Fig S6B). These data indicate a close relationship between the function of SLC25A46, mitochondrial lipid content, cristae formation, and mitochondrial morphology mediated through the oligomerization of MFN2 and OPA1.

## Discussion

This study clearly shows that SLC25A46 is present in discrete foci in the mitochondrial reticulum that co-localize with proteins of the fission/fusion apparatus. Essentially, all fusion and fission events comprise at least one and usually two SLC25A46-positive foci, suggesting that the protein may have an important role in the completion of these events. The observation that deletion of SLC25A46 results in marked mitochondrial fragmentation and that the re-expression of all tested loss-of-function pathogenic variants in the null cell line resulted in a hyperfused mitochondrial reticulum shows that SLC25A46 is not an essential component of the fission/fusion apparatus per se. Mitochondria can still fuse in the complete absence of SLC25A46, and they divide in the presence of the hypomorphic pathogenic variants, which, like the WT protein, also appear to be organized in foci. The mitochondrial hyperfusion phenotype associated with SLC25A46 pathogenic variants is a consistent finding in studies by other groups (Abrams et al, 2015; Janer et al, 2016; Wan et al, 2016; Wang et al, 2022).

These data suggest that the alterations in mitochondrial morphology might reflect cellular stresses caused by the loss of the SLC25A46 function, an idea supported by the striking defects in proliferation that are independent of mitochondrial morphology. Decreased levels of SLC25A46 might induce stress-induced mitochondrial hyperfusion, a phenomenon described in response to nutrient starvation and oxidative stress (Tondera et al, 2009; Shutt et al, 2012; Shutt & McBride, 2013). In contrast, mitochondrial fragmentation is a requirement for mitophagy (Arnoult et al, 2005; Gomes & Scorrano, 2008; Twig et al, 2008). Although the depletion of SLC25A46 in fibroblasts did not result in obvious assembly defects in OXPHOS complexes analyzed by BN-PAGE ([Janer et al, 2016]; Fig S2C), despite the striking abnormalities in cristae ultrastructure, our

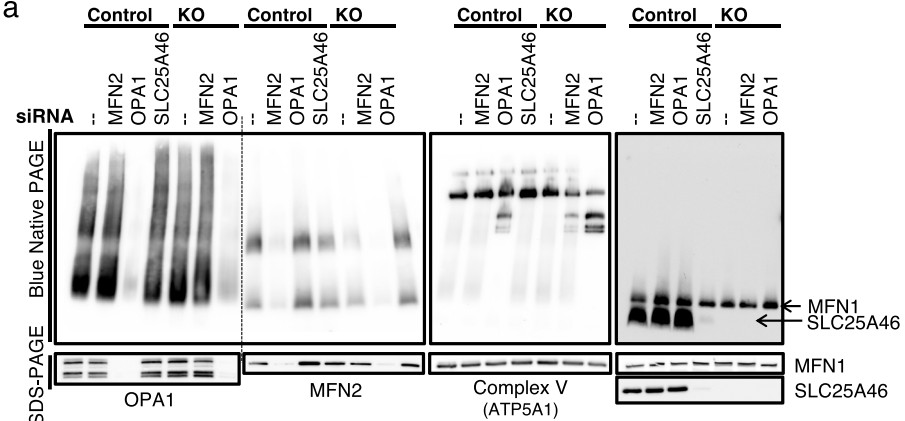

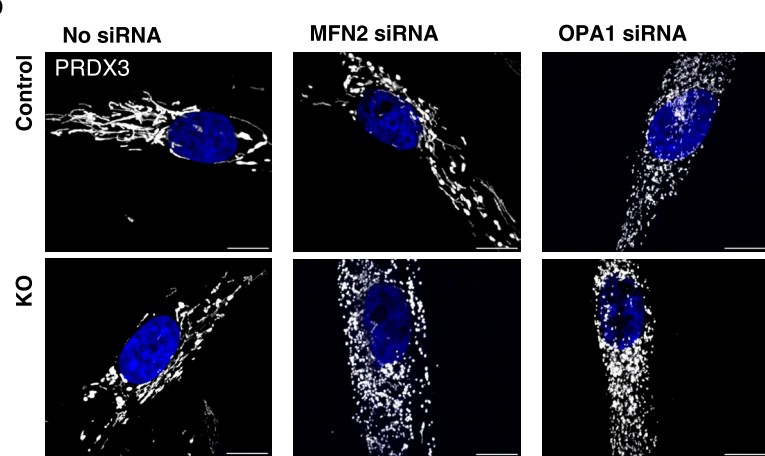

**Figure 7. MFN2 oligomerization is influenced by SLC25A46 and OPA1.**
**(A)** Analysis of MFN2 and OPA1 oligomerization (by BN-PAGE and SDS–PAGE) of isolated mitochondria from control and SLC25A46 knock-out fibroblasts in which MFN2 or OPA1 was knocked down by siRNA constructs for 6 d. siRNA-mediated knock-down of SLC25A46 in control cells is also shown. **(B)** Immunofluorescence analysis of fibroblasts after 6 d of siRNA-mediated knock-down of MFN2 and OPA1 in control and SLC25A46 knock-out fibroblasts. Mitochondria were decorated with anti-PRDX3 in white and DAPI in blue. Scale bar: 10 μm.

previous studies on the T142I variant showed a clear defect in oxygen consumption, perhaps reflecting reduced efficiency of a disorganized OXPHOS system (Janer et al, 2016).

Mitochondrial fusion and fission occur at ER–mitochondrial membrane contact sites. DRP1 and the mitofusins are only recruited after the ER has marked the sites of fission or fusion (Friedman et al, 2011; Abrisch et al, 2020). Mitofusins, DRP1, and OPA1 accumulate in punctae or foci to both fission and fusion sites (Anand et al, 2014; Abrisch et al, 2020), and we show here that SLC25A46 is also present at these sites in proximity to the proteins that are required for fusion and fission (Bleazard et al, 1999; Cipolat et al, 2004). SLC25A46 also co-immunoprecipitates with MFN1, MFN2, and OPA1, consistent with results obtained by BioID proximity mapping. Mitochondrial fission is turning out to be more complex than originally thought involving other organelles including Golgi-derived vesicles and lysosomes (Wong et al, 2018; Nagashima et al, 2020). In our BioID data, we were able to show that proximity interactors of SLC25A46 include proteins of the ER, Golgi vesicles, and other vesicles involved in organellar transport; however, the precise molecular function of these organellar contact sites and vesicles in fission events remains largely unknown.

The oligomerization of both MFN2 and OPA1 was altered by the loss of the SLC25A46 function. High molecular weight oligomers of MFN2 were reduced in the null cell line and in the presence of all three pathogenic variants, a reduction that correlated with the steady-state level of the residual SLC25A46 protein. Thus, rather unexpectedly MFN2 oligomerization did not correlate with mitochondrial morphology in our model. On the contrary, the oligomerization of OPA1 had a clear relationship with mitochondrial morphology.

On an SDS–PAGE, one can distinguish between five different forms of OPA1 for which the long forms are described to be required for the mitochondrial fusion of the inner membranes, whereas the short forms are soluble and are associated with fission (MacVicar & Langer, 2016). Although OPA1 is involved in mitochondrial dynamics, it also has an important role in maintaining the cristae architecture by stabilizing the cristae junctions and determining MICOS assembly (Stephan et al, 2020). The SLC25A46 knock-out cell lines or the expression of the pathogenic variants of SLC25A46 leads to an impaired cristae structure and an increase in the higher molecular weight complexes of OPA1 (>600 kD), which align with the MICOS complex, suggesting that OPA1 might accumulate at the cristae junctions for increased stability. As we observed mitochondrial fragmentation in the SLC25A46 knock-out cells, we saw an increase in the shorter forms of OPA1 in the higher molecular weight complexes. On the contrary, the expression of the pathogenic variant p.T142I results in mitochondrial elongation and an increase in the longer forms of OPA1, suggesting

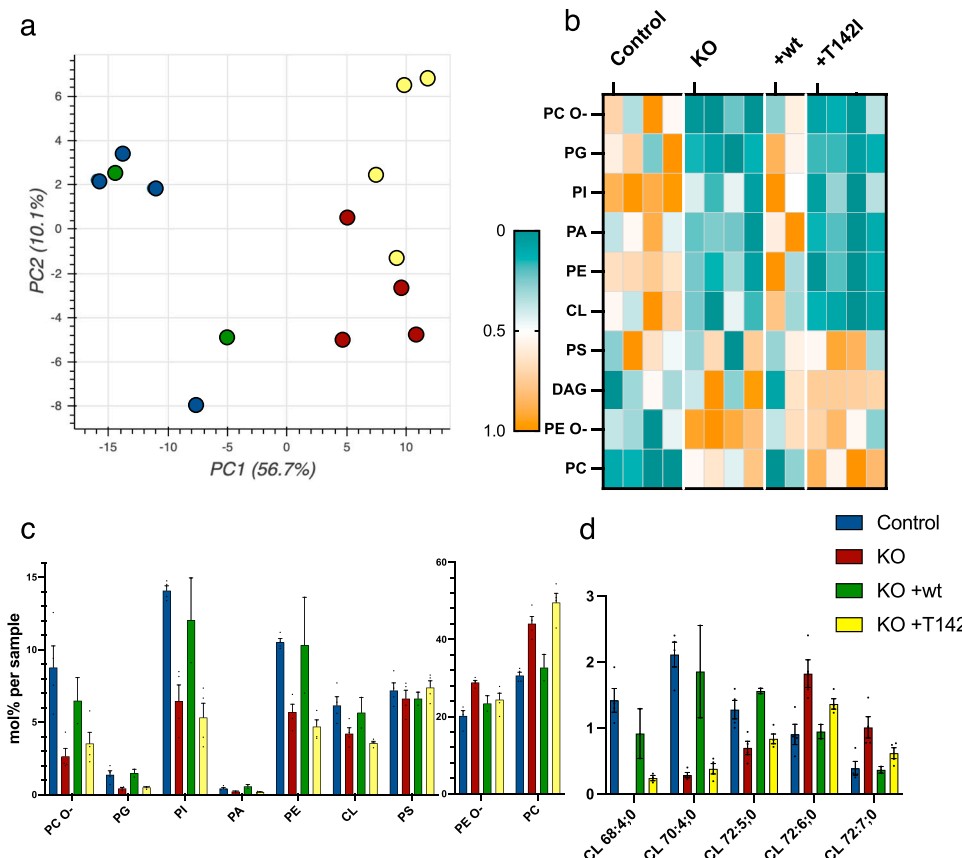

**Figure 8. SLC25A46-dependent changes in the mitochondrial lipid profile.**
Sucrose bilayer purified mitochondria from control human fibroblasts (control, n = 4), SLC25A46 knock-out fibroblasts (KO, n = 4), and SLC25A46 knock-out fibroblasts expressing either the WT protein (KO + wt, n = 2) or the pathogenic variant T142I (KO + T142I, n = 4) were analyzed for absolute quantification of lipid content using shotgun mass spectrometry lipidomics. **(A)** PCA of individual samples. **(B)** Heat map representing the relative abundance of the lipid classes (ether-linked phosphatidylcholine [PC O-], phosphatidylglycerol [PG], phosphatidylinositol [PI], phosphatidic acid [PA], phosphatidylethanolamine [PE], cardiolipin [CL], phosphatidylserine [PS], diacylglycerol [DAG], ether-linked PE [PE O-], and phosphatidylcholine [PC]) of the 203 significantly different lipid entities detected by mass spectrometry. Lipids were sorted by fold change between control and knock-out cell lines. Data were normalized within each class of lipids. **(C)** Lipid class profile represented as molar % of the total lipid amount (mol%). **(D)** Lipid species profile of cardiolipins. Data are presented as molar % of the total lipid amount (mol%).

that the morphology phenotype is driven by the relative abundance of long and short forms of OPA1.

The oligomerization of both GTPases requires a particular lipid environment (Rujiviphat et al, 2009; Ban et al, 2010; Macdonald et al, 2014; Frohman, 2015), and we hypothesize that SLC25A46 may be involved in providing a lipid platform for MFN2 and OPA1 oligomerization. The SLC25A46 homolog Ugo1p (Abrams et al, 2015) in yeast facilitates the dimerization of the MFN1/2 homolog Fzo1p (Anton et al, 2011). The mechanism is unclear, but perhaps it is also involved in creating a lipid environment conducive to GTPase oligomerization. Furthermore, knock-down of OPA1 leads to a rescue of the decreased MFN2 oligomerization in the SLC25A46 knock-out cell line, whereas the phenotype of fragmented mitochondria was not rescued. To our knowledge, oligomers of MFN2 have not been described to correlate with levels of OPA1; however, it is known that OMA1 cleavage of OPA1 is MFN2-dependent—MFN2 depletion leads to an increase in longer forms of OPA1 because of decreased OPA1 cleavage by OMA1 (Sood et al, 2014). We did not observe an increase in longer OPA1 forms when we knocked down MFN2, nor were we able to see a difference in the assembly of OPA1 complexes (Fig 7A).

Mitochondrial membranes have a high content of phospholipids for which they depend on the ER for lipid transfer (Daum & Vance, 1997; Lev, 2012). Although phosphatidic acid (PA), PS, and PC are largely synthesized in the ER, the synthesis of PE and the mitochondrial-specific diglycerophospholipid cardiolipin (CL)

occurs at the mitochondrial inner membrane. The outer and inner mitochondrial membranes differ significantly in the lipid content as CL is enriched in the inner membrane and only present in low concentrations in the outer membrane (de Kroon et al, 1997; Tatsuta et al, 2014). We found a decreased content of the three main mitochondrial lipids (CL, PE, and PI) in the SLC25A46 knock-out, but PC was increased. The synthesis of PC occurs in the ER from mitochondrial-derived PE (Tondera et al, 2009), and therefore, it is unlikely that lipid flux between these two organelles is decreased in the absence of SLC25A46, consistent with the observation of no reduction in ER–mitochondrial contact sites in EM studies (Fig 2E). The role of SLC25A46 might be more related to the organization of a focus for lipid transfer or mixing at sites of fusion and fission. Fusion and fission are facilitated through a membrane potential, which can be generated through specific lipids inducing membrane curvature. Furthermore, fusion and fission require a specific lipid content, which supports the oligomerization of the GTPases (Frohman, 2015). We saw a decrease in the overall class of CL and the different abundance of subspecies of CL in the absence of SLC25A46. It has been shown that the negatively charged CL binds DRP1 and OPA1, which drives activation of the GTPases, inducing fission and fusion, respectively (Rujiviphat et al, 2009; Ban et al, 2010; Macdonald et al, 2014). The altered CL content in the knock-out of SLC25A46 might explain the oligomerization deficit of MFN2 and OPA1. The observation that the levels of several lipid subspecies were significantly different in SLC25A46 knock-out cells compared

with those re-expressing the p.T142I variant may be related to the strikingly different mitochondrial morphology in these cells, but this requires further investigation. Along with the altered lipid content, we showed impaired cristae morphology in the SLC25A46 null cells, and because mitochondrial fusion and fission events require certain lipids and SLC25A46 is present at these events, we speculate that SLC25A46 is involved in facilitating lipid flux to the site of fusion or fission facilitating the oligomerization of the required GTPases for fusion and fission. A recent study on MTCH2, another mitochondrial outer membrane protein in the SLC25 family, showed that it stimulated mitochondrial fusion through an interaction with lysophosphatidic acid (Labbe et al, 2021). This was suggested to be a mechanism of signaling lipogenic flux to the mitochondrial fusion machinery. Perhaps the modified SLC25 carriers that localize to the OMM and do not have the conserved sequences to form a pore for the transport of small metabolites have evolved to monitor or regulate different aspects of lipid transactions between the ER and mitochondria.

# Materials and Methods

## Cell lines

Fibroblasts were obtained from a cell bank located in the Montreal Children's Hospital, and the cell line we used was from a female healthy subject, 58 yr old. The cells were immortalized as described previously (Lochmuller et al, 1999). Fibroblasts, the Phoenix packaging cell line, and Flp-In T-REx 293 cells were cultured in high-glucose DMEM (Wisent 319-027-CL) supplemented with 10% FBS, at 37°C in an atmosphere of 5% $CO_2$. All cell lines were regularly tested for mycoplasma contamination.

## CRISPR/Cas9 SLC25A46 knock-out cell line generation

The sgRNA oligomers for SLC25A46 targeting exons 1 and 3 were annealed and inserted into the plasmid pSpCas9(BB)-2A-Puro (PX459) V2.0 from Addgene (62988) via combined ligation (T7 ligase; NEB).

Sequence for sgRNA targeting exon 1 of SLC25A46:
5'-CTCCGTGGGGCCTTCGTACG GGG-3'
Sequence for sgRNA targeting exon 3 of SLC25A46:
5'-GGCGGCGTAGAACAATGCA AGG-3'

Both plasmids (1.5 μg) were transfected into a fibroblast cell line with Lipofectamine 3000 (Thermo Fisher Scientific) reagent according to the manufacturer's instructions. The day after, transfected cells were selected by addition of puromycin (2.5 μg/ml) for 2 d. Clonal cells were screened for the loss of target protein by immunoblotting, and frameshift mutations were confirmed by genomic sequencing.

## Plasmid generation

SLC25A46 was amplified through PCR with Taq polymerase with 7-deaza-GTP/nucleotide mix (NEB) using cDNA from control fibroblasts as a template and cloned into Gateway-modified pBABE-Puro

using Gateway Cloning Technology (Invitrogen). The SLC25A46-GFP plasmid was generated by cloning SLC25A46 into a pDEST-pcDNA5-GFP vector. The three pathogenic variants were created using the QuikChange Lightning Site-Directed Mutagenesis Kit (Agilent Technologies) with primers:

p.T142I: c.C425T 5'-cagcattaccatctcactccatttatagtcatcaatattatgtaca-3'
p.R257Q: c.G770A 5'-gagtgcctcatagcaaacaacttcttccgcttctttc-3'
p.E335D: c.A1005T 5'-cgttatactttacccattggatacagttttgcaccgcc-3'
pOCT-PAGFP was previously described in Nagashima et al (2020) and obtained from Addgene. The vector was cloned into a pcDNA-pDEST47 Gateway destination vector (Invitrogen) and flipped into a retroviral expression vector pLXSH.

The sequence of all plasmids was verified by Sanger sequencing (IRIC).

## Generating overexpression cell lines

Retroviral constructs were transfected into the Phoenix packaging cell line using the HBS/$Ca_3(PO_4)_2$ method. Control fibroblasts or SLC25A46 knock-out cells were infected 48 h later by exposure to a virus-containing medium in the presence of 4 μg/ml polybrene as described previously (Pear et al, 1997). Finally, antibiotic selection began 2–3 d later with 4 μg/ml of puromycin or 100 U/ml of hygromycin for the pBABE vector or pLXSH vector-infected cells, respectively.

## Neuronal cell culture

Neurons derived from induced pluripotent stem cells were kindly provided by Thomas Durcan's Lab (EDDU, The Neuro). Immunofluorescence analysis of cortical neurons was conducted on day 14.

## siRNA transfection

Stealth RNAi duplex constructs (Invitrogen) were used for transient knock-down of SLC25A46 (S40836; Ambion), and ON-TARGETplus siRNAs were used for OPA1, MFN2, and DRP1 (L-005273, L-012961, and L-012092; Dharmacon) knock-down in fibroblasts. siRNAs were transiently transfected into cells using Lipofectamine RNAiMAX (Invitrogen), according to the manufacturer's specifications. The transfection was repeated on day 3, and the cells were harvested on day 6 for immunofluorescence and immunoblot analyses.

## Preparation of mitochondria-enriched heavy membranes

Mitochondria-enriched heavy membrane fractions were acquired from two 90% confluent 15-cm plates. The plates were washed twice with PBS, and cells were collected and resuspended in HIM buffer (0.2 M mannitol, 0.07 M sucrose, 0.01 M Hepes, and 1 mM EGTA, pH 7.5) and homogenized with 10 passes of a pre-chilled, zero-clearance homogenizer (Kimble/Kontes). The post-nuclear supernatant was obtained by centrifugation of the samples twice for 10 min at 600$g$. The heavy membrane fraction was pelleted by centrifugation for 10 min at 10,000$g$ and washed once in HIM buffer. The protein concentration was measured by the Bradford assay.

## Alkaline carbonate extraction and proteinase K protection assays

To determine the suborganellar location of the pathogenic variants of SLC25A46, mitochondria were extracted with 100 mM alkaline carbonate at pH 11.5 as previously described (Janer et al, 2016) and the input, pellet, and supernatant were analyzed by SDS–PAGE. To determine the localization of the pathogenic variant R257Q at the membrane, 100 μg of freshly prepared crude mitochondria were incubated with an increasing concentration of proteinase K diluted in the isolation buffer, for 20 min on ice. The reaction was stopped by addition of PMSF (2 mM final), and the cells were incubated on ice for 20 min. Mitochondria were then pelleted at 10,000$g$ for 10 min, resuspended in Laemmli buffer, and analyzed by SDS–PAGE.

## Denaturing, native, and second-dimension PAGE

For SDS–PAGE, cells were extracted with 1.5% n-dodecyl-D-maltoside (DDM) in PBS and centrifuged at 20,000$g$ for 20 min, after which 20 μg of protein was run on polyacrylamide gels.

BN-PAGE was used to separate individual protein complexes. Each sample was obtained from 100 μg mitochondria-enriched heavy membrane fractions. Mitochondria were resuspended in 40 μl MB2 buffer (1.75 M aminocaproic acid, 75 mM Bis–Tris, and 2 mM EDTA) and solubilized with 4 μl 10% digitonin solution (4 g digitonin/g protein) for 20 min on ice with reciprocal centrifugation for 20 min at 20,000$g$. 15 μg of supernatants was separated on a 6–15% polyacrylamide gradient gel as previously described (Leary, 2012), with an Amersham HMW native marker kit (17044501; GE Healthcare) as a molecular weight marker, and blotted onto a polyvinylidene difluoride membrane.

For the second-dimension analysis, BN-PAGE/SDS–PAGE was carried out as detailed previously (Antonicka et al, 2003). Separated proteins were transferred to a nitrocellulose membrane, and immunoblot analysis was performed with the indicated antibodies.

## Immunofluorescence

For immunofluorescence analysis, cells were fixed in 4% formaldehyde in PBS at 37°C for 20 min, then washed three times with PBS, followed by permeabilization in 0.1% Triton X-100 in PBS and three washes in PBS. The cells were then blocked with 4% FBS in PBS, followed by incubation with primary antibodies in 4% FBS in PBS, for 1 h at RT. After three washes with PBS, cells were incubated with DAPI and the appropriate anti-species secondary antibodies coupled to Alexa fluorochromes (Invitrogen) (1:3,000) for confocal microscopy or with Abberior STAR antibodies (1:1,000) for STED microscopy for 1 h at RT. After three washes in PBS, coverslips were mounted onto slides using fluorescence mounting medium (Dako).

## Confocal imaging and STED super-resolution microscopy

For confocal microscopy, stained cells were imaged using 100× objective lenses (NA1.4) on an Olympus IX81 inverted microscope with appropriate lasers using an Andor/Yokogawa spinning disk system (CSU-X), with a sCMOS camera. The laser power of 10 (for, i.e., anti-PRDX3, MitoTracker, and anti-OPA1) or 20% (anti-SLC25A46 and SLC25A46-GFP) with a dwell time of 100–500 μs was used,

depending on the strength of the antibody. Mitochondrial network morphology (Fig 2) was classified in a blinded manner as fused, intermediate, or fragmented. For each condition, more than 60 cells were analyzed. Experiments were done three times independently. Errors bars represent the mean ± SEM, and $P$-values were calculated using a $t$ test. The fluorescence of PAGFP-expressing cells was stimulated by excitation with the 405 nm laser of a ~1-μm$^2$ square close to the nucleus. An image was taken 3 min after stimulation. The PAGFP-positive area of OCT-PAGFP–infected cells was obtained using the Fiji (Schindelin et al, 2012) macro, which automatically sets the threshold and uses the "Analyze particles" plugin in Fiji. At least 20 cells per condition were analyzed.

For live-cell imaging analysis, cells were incubated with Mito-Tracker Deep Red FM (Thermo Fisher Scientific) for 15 min before imaging and washed three times. Time-lapse videos were acquired over the course of 1 min with each channel captured every 0.5 s.

STED images were obtained with an Abberior Expert Line STED microscope with two pulsed STED lasers (595 and 775 nm) based on an Olympus IX83 inverted microscope with an Olympus Plan-Apo ×100/1.40 NA oil objective and a pixel size of 20 nm using ImSpector software. The laser power of 90% was used for confocal lasers, and the laser power of 100% was used for the STED laser with dwell times of 5 and 20 μs, respectively.

## TEM

The TEM was performed at the Facility for Electron Microscopy Research of McGill University. Cells were washed in 0.1 M Na-cacodylate washing buffer (Electron Microscopy Sciences) and fixed in 2.5% glutaraldehyde (Electron Microscopy Sciences) in 0.1 M Na-cacodylate buffer overnight at 4°C. Cells were washed three times in 0.1 M Na-cacodylate washing buffer for a total of 1 h, incubated in 1% osmium tetroxide (Mecalab) for 1 h at 4°C, and washed with ddH2O three times for 10 min. Then, dehydration in a graded series of ethanol/deionized water solutions from 30 to 90% for 8 min each, and 100% twice for 10 min each, was performed. The cells were then infiltrated with a 1:1 and 3:1 Epon 812 (Mecalab): ethanol mixture, each for 30 min, followed by 100% Epon 812 for 1 h. Cells were embedded in the culture wells with new 100% Epon 812 and polymerized overnight in an oven at 60°C. Polymerized blocks were trimmed, and 15–20 100-nm ultrathin sections per sample were cut with an UltraCut E ultramicrotome (Reichert Jung) and transferred onto 200-mesh Cu grids (Electron Microscopy Sciences). Sections were post-stained for 8 min with 4% aqueous uranyl acetate (Electron Microscopy Sciences) and 5 min with Reynold's lead citrate (Thermo Fisher Scientific). The samples were imaged with a FEI Tecnai-12 transmission electron microscope (FEI Company) operating at an accelerating voltage of 120 kV equipped with an XR-80C AMT, 8-megapixel CCD camera.

## Immunoprecipitation

Before extraction, mitochondria-enriched heavy membrane fractions (200 μg) from control fibroblasts were chemically cross-linked with 1 mM dithiobis-sulfosuccinimidyl propionate (DSP) (Sigma-Aldrich) in HIM buffer, for 2 h on ice. The reaction was stopped by adding glycine, pH 8.0, at 70 mM final concentration for 10 min on

ice. Mitochondria were pelleted, rinsed once, and extracted in 200 μl of lysis buffer (10 mM Tris–HCl, pH 7.5, 150 mM NaCl, 1% n-dodecyl-D-maltoside [DDM] [Sigma-Aldrich], and complete protease inhibitors [Roche]) on ice for 30 min. The extract was centrifuged at 20,000$g$ at 4°C for 20 min, and the supernatant was pre-cleared overnight with non-coated Dynabeads Protein A (Invitrogen) to reduce non-specific protein binding to the beads. Binding of indicated antibodies to Dynabeads Protein A (Invitrogen) was performed overnight. Antibodies were then cross-linked to the beads using 20 mM dimethyl pimelimidate (DMP) (Sigma-Aldrich). The immunoprecipitation reaction was performed overnight at 4°C. Beads were washed with lysis buffer, and samples were eluted using 0.1 M glycine, pH 2.5/0.5% DDM, precipitated with trichloroacetic acid, and immunoblotted as indicated above.

## Proliferation assay

20,000 cells were seeded in a 24-culture dish at day 0. After 1, 2, 3, and 4 d of culture, cells were trypsinized, homogenized, and counted using a Neubauer counting chamber. Experiments were done in independent triplicates. Errors bars represent the mean ± SEM, and $P$-values were calculated using two-way ANOVA.

## Half-life measurement

Flp-In T-REx 293 cells expressing SLC25A46 or the pathogenic variants with a BirA* tag were created as described earlier (Antonicka et al, 2020). The expression of the construct was induced by addition of 1 μg/ml tetracycline (Sigma-Aldrich) for 24 h. The cells were washed and incubated with medium without tetracycline. The cells were harvested after 0, 6, 9, 12, 15, 18, and 24 h, extracted in 1.5% DDM/PBS, and centrifuged at 20,000$g$ for 20 min, after which 20 μg of protein was run on polyacrylamide gels. The SDS–PAGE was quantified using ImageJ, using the endogenous SLC25A46 expression as a loading control. As the intensity of the signal for BirA*-tagged proteins increased up to 6 h, most likely because of the remaining tetracycline within the cells, the signal at 6 h post-removal of tetracycline was considered as the maximal synthesis rate and was set as time 0 for the half-life measurement. The intensity of the normalized SLC25A46-BirA* signal at this time point was set to 1, to which the signal at the following time points was compared with.

## BioID analysis

BioID data published in Antonicka et al (2020) were used for the determination of the specificity of the SLC25A46 proximity inter-actome. The data were loaded into ProHits-viz software (Knight et al, 2017), and a specificity analysis tool was used (BFDR < 0.01; prey-length–normalized average spectral count was used as an abundance measure). The functional enrichment analysis was performed using g:Profiler (version e106_eg53_p16_65fcd97) with the g:SCS multiple testing correction method with a significance threshold of 0.05 (Raudvere et al, 2019) as part of the ProHits-viz analysis platform (Knight et al, 2017).

## Antibodies

Antibodies directed against the following proteins were used in this study: SLC25A46 (G2-SC-515823; Santa Cruz), MFN2 (11925S; Cell Signaling), MFN1 (14739S; Cell Signaling), OPA1 (67589S; Cell Signaling) for IF, OPA1 (612607; BD Bioscience) for WB, DRP1/DLP1 (611113; BD Bioscience), VDAC1 (ab14734; Abcam), CHCHD3/MIC19 (ARP57040-P050; Aviva), CHCHD6/MIC25 (20638-1AP; Proteintech), PRDX3 (in-house), NDUFA9 (ab14713; Abcam), ATP5A1 (ab14748; Abcam), Core1 (ab110252; Abcam), COX4I1 (in-house), and SDHA (ab14715; Abcam); the secondary antibodies used in this study were as follows: anti-rabbit Alexa Fluor 488 (A-21206; Invitrogen), anti-mouse Alexa Fluor 647 (A-31571; Invitrogen), anti-rat Alexa Fluor 594 (A-11007; Invitrogen), Abberior STAR 580 goat anti-rabbit IgG (ST580-1002; STED), and Abberior STAR 635P goat (2-0002-007-5; Abberior GmbH).

## Mitochondrial isolation for lipidomics

To obtain at least 100 μg of isolated mitochondria from fibroblasts, 4 × 15 cm plates with 90% confluency were grown for each sample. The plates were rinsed twice with PBS, resuspended in ice-cold 250 mM sucrose/10 mM Tris–HCl (pH 7.4), and homogenized with 10 passes of a pre-chilled, zero-clearance homogenizer (Kimble/ Kontes). A post-nuclear supernatant was obtained by centrifugation of the samples twice for 10 min at 600$g$. The heavy membrane fraction was pelleted by centrifugation for 10 min at 10,000$g$ and washed once in the same buffer. The resuspended membranes were loaded on a 1.7 and 1 M sucrose bilayer and centrifuged in an ultracentrifuge at 75,000$g$ for 30 min. The layer between the 1.7 and 1 M sucrose layers was collected and washed in the 250 mM sucrose/10 mM Tris–HCl buffer. The protein concentration was determined by the Bradford assay.

## Lipidomics analysis by the Lipotype Shotgun Lipidomics technology

Purified mitochondria were used for lipidomics analysis by the Lipotype Shotgun Lipidomics platform. The samples were extracted and analyzed through high-resolution Orbitrap mass spectrometry and quantified and identified by Lipotype's in-house software LipotypeXplorer.

## Statistical analysis

All data are reported as means ± SEM or means ± SD as indicated in the figure legend. Statistical significance was determined using the indicated tests. $P$-values < 0.05 were considered statistically significant and labeled as follows: *$P$ < 0.05, **$P$ < 0.01, and ***$P$ < 0.001.

# Data Availability

The GO term functional analysis "Biological Processes" of the SLC25A46 BioID data of Fig 4 are available in Table S1. The full data of the mass spectrometry lipidomics analysis of Figs 8 and S6 are

deposited in Table S2. The source data of the Western blots shown in Figs 1 and 5 are provided.

## Supplementary Information

## Acknowledgements

This research was supported by a grant from the CIHR to EA Shoubridge (FRN178373). We thank the Facility for Electron Microscopy Research (FEMR) of McGill University for the electron microscopy sample preparation. We thank Thomas Stroh of the MNI Microscopy Unit for access and assistance with the STED microscope. We thank Gilles Maussion and Thomas Durcan from The Neuro's Early Drug Discovery Unit for providing us with iPSC-derived neurons. We thank Michiel Krols for his expertise with ImageJ macros and Heidi McBride for her expertise and fruitful discussion.

### Author Contributions

J Schuettpelz: conceptualization, data curation, formal analysis, investigation, methodology, and writing—original draft.
A Janer: conceptualization, supervision, investigation, and writing—review and editing.
H Antonicka: conceptualization, investigation, and writing—review and editing.
EA Shoubridge: conceptualization, supervision, and writing—original draft.

### Conflict of Interest Statement

The authors declare that they have no conflict of interest.

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
