## [Reviewer comments · Life Science Alliance]

Life Science Alliance

The role of the mitochondrial outer membrane protein SLC25A46 in mitochondrial fission and fusion

Jana Schuettpelz, Alexandre Janer, Hana Antonicka, and Eric Shoubridge

DOI: <https://doi.org/10.26508/lsa.202301914>

Corresponding author(s): Eric Shoubridge, McGill University

Review Timeline:

Submission Date:	2023-01-09
Editorial Decision:	2023-02-20
Revision Received:	2023-03-08
Accepted:	2023-03-10

Scientific Editor: Novella Guidi

Transaction Report:

Please note that the manuscript was reviewed at Review Commons and these reports were taken into account in the decision-making process at Life Science Alliance.

Revision Plan

Manuscript number: RC-2022-01673

Corresponding author(s): Eric Shoubridge

[The “revision plan” should delineate the revisions that authors intend to carry out in response to the points raised by the referees. It also provides the authors with the opportunity to explain their view of the paper and of the referee reports.]

The document is important for the editors of affiliate journals when they make a first decision on the transferred manuscript. It will also be useful to readers of the reprint and help them to obtain a balanced view of the paper.

*If you wish to submit a full revision, please use our "Full Revision" template. **It is important to use the appropriate template to clearly inform the editors of your intentions.**]*

1. General Statements [optional]

This section is optional. Insert here any general statements you wish to make about the goal of the study or about the reviews.

2. Description of the planned revisions

To answer the comments of Reviewer #1 we will need two more weeks:

- 1. We have run WB- and new BN-PAGEs to visualize DRP1 and several DRP1 receptors. We are in the process of repeating these experiments to ensure that the results are reproducible.*
- 2. The reviewer asked for the localization of the pathogenic variants of SLC25A46. We performed a protease protection assay for the pathogenic variant R257Q (most abundant) which demonstrated that it localized to the outer membrane as does the wild-type protein. In the coming week, we will perform alkaline carbonate extractions for all pathogenic variants to test whether all are integral membrane proteins (which we predict)*
- 3. The reviewer is uncertain about the sizes of the complexes. We run every BN-PAGE with a ladder. We will however repeat these experiments by blotting with antibodies against all the OXPHOS complexes as additional support for the for the molecular masses of the MFN2 and OPA1 complexes.*
- 4. We can answer this with a written statement*
- 5. We have already quantified the data*

To answer the questions of Reviewer 2 we have already prepared our written responses

Revision Plan

3. Description of the revisions that have already been incorporated in the transferred manuscript

Please insert a point-by-point reply describing the revisions that were already carried out and included in the transferred manuscript. If no revisions have been carried out yet, please leave this section empty.

4. Description of analyses that authors prefer not to carry out

Please include a point-by-point response explaining why some of the requested data or additional analyses might not be necessary or cannot be provided within the scope of a revision. This can be due to time or resource limitations or in case of disagreement about the necessity of such additional data given the scope of the study. Please leave empty if not applicable.

Manuscript number: RC-2022-01673

Corresponding author(s): Eric Shoubridge

[Please use this template only if the submitted manuscript should be considered by the affiliate journal as a full revision in response to the points raised by the reviewers.]

*If you wish to submit a preliminary revision with a revision plan, please use our "Revision Plan" template. **It is important to use the appropriate template to clearly inform the editors of your intentions.**]*

1. General Statements [optional]

This section is optional. Insert here any general statements you wish to make about the goal of the study or about the reviews.

This section is mandatory. Please insert a point-by-point reply describing the revisions that were already carried out and included in the transferred manuscript.

Reviewer #1 (Evidence, reproducibility and clarity (Required)):

In this MS, Scheuttpelz et al demonstrate that SLC25A46, a novel member of the mitochondrial carrier protein family localized to the outer-mitochondrial membrane, is an important regulator of mitochondrial dynamics. They show that knockout of SLC25A46 results in mitochondrial fragmentation, whereas over-expression of WT SLC25A46 or pathogenic variants/mutants of SLC25A46 results in mitochondria hyperfusion. SLC25A46 might affect fusion/fission directly since it is localized to both mitochondrial fusion and fission sites. Moreover, its loss/expression of variants alters the levels of the high molecular weight complexes of MFN2 and alters the levels of the long/short forms of OPA1. In addition, Scheuttpelz et al show that loss of SLC25A46 results in changes in the mitochondrial lipid profile, suggesting that SLC25A46 might regulate mitochondrial dynamics via regulation of mitochondrial lipid metabolism. Thus, the findings described are novel and exciting, however it remains poorly understood how SLC25A46 localization to fusion/fission sites is related to mitochondrial fusion/fission, and how are these results related to its effect on the MFN2/OPA1 complexes/forms, and to its possible role in regulating lipid metabolism.

Specific comments:

- 1. Are the DRP1 and the DRP1-receptor native complexes (appearing in BN-PAGE) altered in the SLC25A46 KO/+pathogenic variants cells?*

We have tried to visualize the DRP-1 receptor complexes (MID49, MID51, MFF) on BN-PAGE gels without success. The Western blot in (a) below shows that the steady-state levels of all three receptors are similar under all conditions we tested, but the same antibodies used in this blot did not detect the native complexes on BN-PAGE gels. To our knowledge this has not been done in the literature. We previously reported (Janer et al, 2016) that DRP1 recruitment in patient fibroblasts (T142I) was only slightly reduced and that its oligomerization state (after crosslinking analysis) was slightly increased in the patient cells, which would not explain the mitochondrial hyperfusion.

2. Do the pathogenic variants of *SLC25A46* localize only to mitochondria? Do they fold similar to the WT protein (i.e., similar prot K cleavage products)? Are they loss- or gain-of-function variants/mutants?

We previously provided images of all pathogenic variants in Supplementary Figure 1 by decorating with an *SLC25A46* antibody; however the low steady-state levels of all but the R257Q variant make visualization difficult. Supplementary Figure 3d shows the R257Q variant with an analysis of its suborganellar localization. We performed a PK assay of R257Q (the most abundant pathogenic variant) and it behaves as the wild-type protein (rescued in knock-out background and in the control cell line) and as the outer membrane protein MFN2. We have now performed an alkaline carbonate extraction assays showing that all pathogenic variants (T142I, R257Q and E335D) are integral membrane proteins. (results shown below)

All described *SLC25A46* mutations are loss-of-function biallelic missense, STOP or frameshift mutations, and where it has been investigated, all are associated with reduced steady-state levels of *SLC25A46* protein compared to controls. The level of residual *SLC25A46* protein correlates with disease severity Abrams et al. (2018).

Proteinase K Assay and Alkaline Carbonate Extraction show an integral insertion of SLC25A46 and its pathogenic variants into the outer membrane.

a) Proteinase K digestion assay of mitochondria from control fibroblasts or SLC25A46 knock-out fibroblasts with reintroduced wild-type protein (+wt) of SLC25A46 or the pathogenic variant R257Q. Mitochondria were exposed to an increasing concentration of proteinase K to determine the submitochondrial localization of SLC25A46. SLC25A46 and its pathogenic variants behave as outer membrane proteins. MFN2 was used as a control for an outer membrane protein, AIF for protein present in the inter-membrane space, and SCO1 for an inner membrane protein.

b) Alkaline carbonate extraction of mitochondria from control fibroblasts or SLC25A46 knock-out fibroblasts with reintroduced wild-type protein (+wt) of SLC25A46 or the pathogenic variants (+T142I, +R257Q, +E335D). Immunoblot analysis shows that all SLC25A46 variants behave as integral membrane proteins. PRDX3 (soluble mitochondrial matrix protein) and MFN2 (integral outer membrane protein) were used as controls.

3. *The BN-PAGE results presented in Fig 5 appear without molecular weight markers, and thus the sizes of the complexes are not known. Why did the authors conclude that the bands that appear in the MFN1, MFN2, and OPA1 blots represent monomers and oligomers of these proteins (Fig 5b)? Is it possible that all/part of these immune-reactive bands represent complexes with other proteins and not monomers and/or homo-oligomers? How does SLC25A46 affect the complex state of these proteins if it does not associate with them in the native state, as seen in Fig 5d?*

We added a molecular weight ladder in Figure 5b which was confirmed using the known molecular weights the complexes of the oxidative phosphorylation complexes.

4. *Fig 5b (MFN2 blot): SLC25A46 KO cells expressing each of the pathogenic variants/mutants of SLC25A46 show different levels of the MFN2-immuoreactive higher molecular weight band (MFN2-HMWB; last three lanes), however all three cell lines show mitochondria hyperfusion. Moreover, the intensity of the MFN2-HMWB in two of these mutant lines (+T142I and +E335D) is similar to the intensity of the band that appears in the SLC25A46 KO cells, cells which show fragmented mitochondria.*

Thus, there is not a clear correlation between the state of SLC25A46, the levels of the MFN2-HMWB, and the mitochondrial morphology.

The reviewer is correct and in fact we discussed this point in the fourth paragraph of the discussion part in our paper: “*The oligomerization of both MFN2 and OPA1 was altered by the loss of SLC25A46 function. High molecular weight oligomers of MFN2 were reduced in the null cell line and in the presence of all three pathogenic variants, a reduction that correlated with the steady-state level of residual SLC25A46 protein. Thus, rather unexpectedly MFN2 oligomerization did not correlate with mitochondrial morphology in our model.*” It thus appears that the oligomerization state of MFN2 is not the determining factor for the observed changes in mitochondrial morphology.

- 5. The authors' interpretations of the results presented in Fig 5d, arguing that there is a correlation between the appearances of the short/long forms of OPA1 and the fusion/fission state of the different cells, are not convincing. BN-PAGE results can vary between experiments, and thus need to be repeated and accompanied by densitometry analyses, especially in cases where the intensity of the bands (short and long forms of OPA1) seem largely similar in the single experiment presented.*

We have now performed additional two-dimensional electrophoresis (BN-PAGE/SDS-PAGE) analyses and have quantified the results. (a) Mitochondria from control, knock-out, re-expression of wt-SLC25A46 and the pathogenic variant p.T142I were run on a BN-PAGE with additional SDS gel-electrophoreses and immunoblotted against OPA1. (b) Quantification of the high molecular weight complexes (>600 kDa) of OPA1 (indicated in the green boxes in (a) relative to the total signal. (c) Quantification of the relative proportions of the long vs short forms of OPA1 forms in the high molecular weight complexes (>200 kDa) as indicated in the example (d) showing the longer forms of the higher complexes in the yellow box and the shorter forms in the purple box.

OPA1 forms high molecular oligomeric complexes that are altered in SLC25A46 loss of function cells

Reviewer #2 (Significance (Required)):

The manuscript "SLC25A46 localizes to sites of mitochondrial fission and fusion and loss of function variants alter the oligomerization states of MFN2 and OPA1" partially characterizes the outer mitochondrial membrane protein SLC25A46, finding a localization to the tips and branching points of mitochondria and an effect on both mitochondrial internal structure and mitochondrial network dynamics in deletions and expression of specific mutants. The localization was conducted both with tagged protein and antibodies, which is appreciated, as tagging and overexpression can often alter localization of mitochondrial proteins. Interestingly, disease variants have an opposite effect as the deletion in mitochondria network behavior, with fragmented mitochondria in deletion strains and elongated or fused mitochondria in the mutant strains. The paper also finds alteration on membrane composition, and postulates a function in lipid exchange. While the paper falls short of a full functional characterization, the results are reasonable, internally consistent, and promising for future follow-ups.

Altered protein expression levels for the disease variant proteins is somewhat of a concern regarding the results, as it can be difficult to parse what cellular effects are due to altered

Full Revision

protein activity versus altered protein levels, however this protein expression effect is consistent with previous literature and is likely unavoidable for this investigation.

Overall, the characterization of SLC25A46's localization, interactions, and effects on protein and mitochondrial structural/network organization suggests a function in mitochondrial OMM contact sites and that loss or mutation of this protein results in significant stress to the mitochondria with downstream effects.

Minor comments:

- What type of fibroblasts were used and was any subject information worth mentioning? I did not find this mentioned anywhere.

We added an explanation in the Materials and Methods: "Fibroblasts were obtained from a cell bank located in the Montreal Children's Hospital and the cell line we used was from a female healthy subject, 58 years old."

- For the confocal and STED microscopy use, what laser power was used for each excitation? More detail on the settings used for imaging with the microscopes would be help for experimental reproducibility.

We have added to the Materials and Methods: For confocal microscopy "A laser power of 10 (for i.e. anti-PRDX3, MitoTracker, anti-OPA1) or 20% (anti-SLC25A46, SLC25A46-GFP) with a dwell time of 100 - 500 μ s was used, depending on the strength of the antibody." For the STED microscopy, we added: "A laser power of 90% was used for the confocal lasers and a laser power of 100% was used for the STED laser with dwell times of 5 μ s and 20 μ s, respectively."

- Figure 7 C - the bar graph is very squished; one can barely see the levels of the small bars.

We have modified Figure 7C to make the results more visible.

a

b

c

d

February 20, 2023

RE: Life Science Alliance Manuscript #LSA-2023-01914-T

Prof. Eric A. Shoubridge
McGill University
Montreal Neurological Institute
& Dept. of Human Genetics
McGill University
Montreal, 3801 University Street H3A 2B4
Canada

Dear Dr. Shoubridge,

Thank you for submitting your revised manuscript entitled "SLC25A46 localizes to sites of mitochondrial fission and fusion and loss of function variants alter the oligomerization states of MFN2 and OPA1". We would be happy to publish your paper in Life Science Alliance pending final revisions necessary to meet our formatting guidelines.

- please address the final Reviewer 2's comments
- please upload both your main and your supplementary figures as single files
- please upload your video file
- please upload your table files as editable doc or excel files
- please add your figure legends (main figures, supplementary figures, video, and tables) as a separate section to your manuscript
- please add a summary blurb/alternate abstract and a category for your manuscript to our system
- please add the Twitter handle of your host institute/organization as well as your own or/and one of the authors in our system
- please consult our manuscript preparation guidelines <https://www.life-science-alliance.org/manuscript-prep> and make sure your manuscript sections are in the correct order
- please use the [10 author names, et al.] format in your references (i.e. limit the author names to the first 10)
- please add a figure callout for Figure 6D
- please double-check your callouts for Figure 8; you have a callout for Figure 8d, but this isn't in the legend or the figure

Figure Check:

- please add the panel (a) to your Figure 8 figure legend
- *Figure 1C: blots look as if they've separately been pasted in; same with figure 5a; please provide source data for these figures

A. FINAL FILES:

- An editable version of the final text (.DOC or .DOCX) is needed for copyediting (no PDFs).
- High-resolution figure, supplementary figure and video files uploaded as individual files: See our detailed guidelines for

preparing your production-ready images, <https://www.life-science-alliance.org/authors>

B. MANUSCRIPT ORGANIZATION AND FORMATTING:

Sincerely,

Reviewer #1 (Comments to the Authors (Required)):

The authors have performed an excellent revision, adequately addressing all my comments, and thus the MS can be accepted for publication

Reviewer #2 (Comments to the Authors (Required)):

The responses to reviewer #1 comment #1-2 look fine to me. I'm not sure about the ladder notation they used in response to comment #3 (strange and limited weights), but I am fine with the expected weight matching the expected complexes. I find the data in response to comment #5 noisy, but I am unclear whether it is statistically significant. If is the case, I think the author has responded to the criticisms.

Reviewer #2 (Comments to the Authors (Required)):

The responses to reviewer #1 comment #1-2 look fine to me. I'm not sure about the ladder notation they used in response to comment #3 (strange and limited weights), but I am fine with the expected weight matching the expected complexes. I find the data in response to comment #5 noisy, but I am unclear whether it is statistically significant. If is the case, I think the author has responded to the criticisms. -please upload both your main and your supplementary figures as single files

It is a Blue Native gel and the ladder represents the following:

Amersham HMW native marker kit (17044501; GE Healthcare): Components • Protein mixture 250 µg/vial, 10 vials contains the following proteins: • Thyroglobulin (1), porcine thyroid, 76 µg, molecular weight (Mr) 669 000 • Ferritin (2), equine spleen, 50 µg, Mr 440 000 • Catalase (3), bovine liver, 36 µg, Mr 250 000 • Lactate dehydrogenase (4), bovine heart, 48 µg, Mr 140 000. • Albumin (5), bovine serum, 40 µg, Mr 66 000. The amount of each protein has been chosen to give bands of equal intensity when stained with Coomassie™ Brilliant Blue following electrophoresis. Intensities may vary when using other staining methods.

We have confirmed the size of our ladder with the known molecular weights of the oxidative phosphorylation complexes as requested by reviewer 1. This is in the revised manuscript.

We had performed additional two-dimensional electrophoresis (BN-PAGE/SDS-PAGE) analyses and have quantified the results. To our knowledge quantifications of second dimensions are not usually performed. We managed to do it and our data are significant, as indicated in the figure legend.

March 10, 2023

RE: Life Science Alliance Manuscript #LSA-2023-01914-TR

Prof. Eric A. Shoubridge
McGill University
Montreal Neurological Institute
& Dept. of Human Genetics
McGill University
Montreal, 3801 University Street H3A 2B4
Canada

Dear Dr. Shoubridge,

Thank you for submitting your Research Article entitled "The role of the mitochondrial outer membrane protein SLC25A46 in mitochondrial fission and fusion". It is a pleasure to let you know that your manuscript is now accepted for publication in Life Science Alliance. Congratulations on this interesting work.

DISTRIBUTION OF MATERIALS:

Again, congratulations on a very nice paper. I hope you found the review process to be constructive and are pleased with how the manuscript was handled editorially. We look forward to future exciting submissions from your lab.

Sincerely,
